# COUNTERFACTUAL THINKING FOR LONG-TAILED INFORMATION EXTRACTION

## ABSTRACT

Information Extraction (IE) aims to extract structured information from unstructured texts. However, in practice, the long-tailed and imbalanced data may lead to severe bias issues for deep learning models, due to very few training instances available for the tail classes. Existing works are mainly from computer vision society, leveraging re-balancing, decoupling, transfer learning and causal inference to address this problem on image classification and scene graph generation. However, these approaches may not achieve good performance on textual data, which involves complex language structures that have been proven crucial for the IE tasks. To this end, we propose a novel framework (named `CFIE`) based on language structure and causal reasoning with three key ingredients. First, by fusing the syntax information to various structured causal models for mainstream IE tasks including relation extraction (RE), named entity recognition (NER), and event detection (ED), our approach is able to learn the direct effect for classification from an imbalanced dataset. Second, counterfactuals are generated based on an explicit language structure to better calculate the direct effect during the inference stage. Third, we propose a flexible debiasing approach for more robust prediction during the inference stage. Experimental results on three IE tasks across five public datasets show that our model significantly outperforms the state-of-the-arts by a large margin in terms of Mean Recall and Macro F1, achieving a relative 30% improvement in Mean Recall for 7 tail classes on the ACE2005 dataset. We also discuss some interesting findings based on our observations.

## 1 INTRODUCTION

The goal of Information Extraction (IE) (Sarawagi, 2008; Chiticariu et al., 2013) is to detect the structured information from unstructured texts. IE tasks, such as named entity recognition (NER) (Lample et al., 2016), relation extraction (RE) (Zeng et al., 2014; Peng et al., 2017) and event detection (ED) (Nguyen & Grishman, 2015) have developed rapidly with the data-hungry deep learning models trained on a large amount of data. However, in real-world settings, unstructured texts follow a long-tailed distribution (Doddington et al., 2004), leading to a significant performance drop on the instance-scarce (or tail) classes which have very few instances available. For example, in the ACE2005 (Doddington et al., 2004) dataset, nearly 70% of event triggers are long-tailed while they only take up 20% of training data. On a strong baseline (Jie & Lu, 2019), the macro F1 score of instance-rich (or head) classes can be 71.6, while the score of tail classes sharply drops to 41.7.

The underlying causes for the above issues are the biased statistical dependencies and spurious correlations between feature representations and classes learned from an imbalanced dataset. For example, an entity *Gardens* appears 13 times in the training set of OntoNotes5.0 (Pradhan et al., 2013), with the NER tag *LOC*, and only 2 times as organization *ORG*. A classifier trained on this dataset will build a spurious correlations between *Gardens* and *LOC*. As a result, an organization that contains the entity *Gardens* may be wrongly predicted as a location *LOC*.

There are only a few studies (Zhang et al., 2019; Han et al., 2018) in the Natural Language Processing (NLP) field to address such long-tailed issues. These works mostly rely on external and pre-constructed knowledge graphs, providing useful data-specific prior information which may not be available for other datasets. On the other hand, there are plenty of works from the computer vision society, where the bias is also quite straightforward. Current solutions include re-balanced training

(Lin et al., 2017) that re-balances the contribution of each class in the training stage, transfer learning (Liu et al., 2019b) that takes advantage of the knowledge in data-rich class to boost the performance of instance-scarce classes, decoupling (Kang et al., 2019) strategy that learns the representations and classifiers separately, and causal inference (Tang et al., 2020a;b; Abbasnejad et al., 2020) that relies on structured causal models for unbiased scene graph generation, image classification and visual question answering.

The aforementioned studies from the computer vision community may not achieve good performance on the textual datasets in the NLP area due to a significant difference between the two fields. For example, unlike images, texts involve complex language structures such as dependency tree and constituent tree that describe the syntactic or semantic level relations between tokens. For the long-tailed IE, how to explore the rich relational information as well as complex long-distance interactions among words as conveyed by such linguistic structures remains an open challenge. Furthermore, to capture a more informative context, the way of utilizing the syntax tree for three IE tasks varies: the RE task relies more on the context and entity type rather than entities themselves, while classifications in NER and ED tasks count more on entities than the context. Hence, it is challenging to decide properly on how to utilize language structures for the above three different IE tasks. One may also think that the prevalent pre-trained models such as BERT (Devlin et al., 2019) may address the long-tailed issues. However, we empirically show that such models still suffer from bias issues.

In this paper, we propose `CFIE`, a novel framework that combines the language structure and counterfactual analysis in causal inference (Pearl et al., 2016) to alleviate the spurious correlations of the IE tasks including NER, RE and ED. From a causal perspective, counterfactuals (Bottou et al., 2013; Abbasnejad et al., 2020) state the results of the outcome if certain factors had been different. This concept entails a hypothetical scenario where the values in the causal graph can be altered to study the effect of the factor. Intuitively, the factor that yields the most significant changes in model predictions have the greatest impact and is therefore considered as main effect. Other factors with minor changes are categorized as side effects. In the context of IE with complex language structures, counterfactual analysis answers the question on "which tokens in the text would be the key clues for RE, NER or ED that could change the prediction result?". With that in mind, our `CFIE` is proposed to explore the language structure to eliminate the bias caused by the side effect and maintain the main effect for the classification. We evaluate our model on five public datasets across three IE tasks, and achieve significant performance gain on instance-scarce classes. We will release our code to contribute the community. Our major contributions are summarized as:

- To the best of our knowledge, our `CFIE` is the first attempt that marries the counterfactual analysis and language structure to address the long-tailed IE issues. We build different structured causal models (SCMs) (Pearl et al., 2016) for the IE tasks and fuse the dependency structure to the models to better capture the main causality for the classification.

- We generate counterfactuals based on syntax structure, where the counterfactuals can be used as interventions to alleviate spurious corrections on models. In doing so, the main effect can be better estimated by the intervention methodology.

- We also propose flexible classification debiasing approaches inspired by Total Direct Effect (TDE) in causal inference. Our proposed approach is able to make a good balance between the direct effect and counterfactuals representation to achieve more robust predictions.

## 2 RELATED WORK

**Long-tailed Information Extraction:** Information extraction tasks, such as relation extraction (Zeng et al., 2014; Peng et al., 2017; Quirk & Poon, 2017), named entity recognition (Lample et al., 2016; Chiu & Nichols, 2016), and event extraction (Nguyen & Grishman, 2015; Huang et al., 2018) are fundamental NLP tasks and have been extensively studied in recent years, For the long-tailed IE, recent models (Lei et al., 2018; Zhang et al., 2019) leverage external rules or transfer knowledge from data-rich classes to the tail classes. Few-shot leaning (Gao et al., 2019; Obamuyide & Vlachos, 2019) has been also applied to IE tasks, although this task focuses more on new classification tasks with only a handful of training instances.

**Re-balancing/Decoupling Models:** Re-balancing approaches include re-sampling strategies (Mahajan et al., 2018; Wang et al., 2020a) that aim to alleviate statistical bias from head classes, and

re-weighting approaches (Milletari et al., 2016; Lin et al., 2017) which assign balanced weights to the losses of training samples from each class to boost the discriminability via robust classifier decision boundaries. These techniques may inevitably suffer the under-fitting/over-fitting issue to head/tail classes (Tang et al., 2020a). There are also recent studies (Kang et al., 2019) that decouple the representation learning and the classifier, which effectively mitigate the performance loss caused by direct re-sampling.

**Casual Inference:** Causal inference (Pearl et al., 2016; Rubin, 2019) and counterfactuals have been widely used in psychology, politics and epidemiology for years. There are many studies in computer vision society (Tang et al., 2020b; Abbasnejad et al., 2020; Tang et al., 2020a; Niu et al., 2020; Yang et al., 2020; Zhang et al., 2020; Yue et al., 2020), which use Total Direct Effect (TDE) analysis framework and counterfactuals for Scene Graph Generation (SGG), visual question answering, and image classifications. There is also a recent work (Zeng et al., 2020) that generates counterfactuals for weakly-supervised NER by replacing the target entity with another entity. Our methods differ from the previous works in three aspects: 1) We explore the syntax structures of texts for building different causal graphs, 2) Counterfactuals are generated based on a task-specific pruned dependency tree. 3) Our proposed inference method yields robust predictions for the NER and ED tasks.

**Model Interpretation:** Besides causal inference, there have been plenty of studies (Molnar, 2020) about traditional model interpretation applied in various applications, such as text and image classification (Ribeiro et al., 2016; Ebrahimi et al., 2018), question answering (Feng et al., 2018; Ribeiro et al., 2018), and machine translation (Doshi-Velez & Kim, 2017). LIME (Ribeiro et al., 2016) was proposed to select a set of instances to explain the predictions. The input reduction method (Feng et al., 2018) is able to find out the most important features and use very few words to obtain the same prediction. Unlike the LIME and input reduction method, the word selections in our `CFIE` are based on the syntax structure. SEARs (Ribeiro et al., 2018) induces adversaries by data augmentation during the training phase. Along this line, a recent study (Kaushik et al., 2019) also uses data augmentation technqiue to provide extra training signal. Our CFIE is orthogonal to data augmenation as it generates counterfactuals during the inference stage, where the counterfactuals are used to mitigate the spurious correlations rather than training the network parameters.

## 3 MODEL

Figure 1 shows the work flow of our proposed `CFIE`. We detail these components as follows.

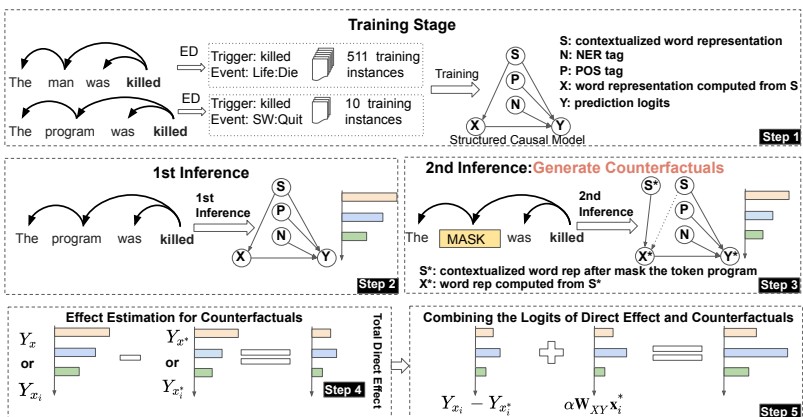

Figure 1: Work flow of our `CFIE` in five steps.

### 3.1 STEP1: CAUSAL REPRESENTATION LEARNING

In this step, we train a causal graph on an imbalanced dataset. Our goal here is to teach the model to identify the main cause (main effect) and the spurious correlations (side effect) for the classification.

**Structural Causal Models (SCMs):** The two well-known causal inference frameworks are SCMs and potential outcomes (Rubin, 2019) which are complementary and theoretically connected. We choose SCMs in our case due to their advantages in expressing and reasoning about the effects of

causal relationships among variables. An SCM can be represented as a directed acyclic graph (DAG) $\mathcal{G} = \{\mathbb{V}, \mathbb{F}, U\}$, where we denote the set of observables (vertices) as $\mathbb{V} = \{V_1, ..., V_n\}$, the set of functions (directed edges) as $\mathbb{F} = \{f_1, ..., f_n\}$, and the set of exogenous variables (e.g. noise) as $U = \{U_1, ..., U_n\}$. Note that in the deterministic case where $U$ is given, the value of all variables in the SCM are uniquely determined (Pearl, 2009). Each observable $V_i$ can be derived from:

$$V_i := f_i(\mathbf{PA}_i, U_i), (i = 1, ..., n),\tag{1}$$

$\forall i, \mathbf{PA}_i \subseteq \mathbf{V}\backslash V_i$ is the set of parents of $V_i$. Directed edges, such as $\mathbf{PA}_i \to V_i$ in the graph $\mathcal{G}$, i.e., $f_i$, refers to the direct causation from the parental variables $\mathbf{PA}_i$ to the child variable $V_i$.

**Our Proposed SCMs:** Figure 2(a) demonstrates our unified SCMs for IE tasks, which are built based on our prior knowledge for the tasks. The variable $S$ indicates the contextualized representations of an unstructured input sentence, where the representations are the output from a BiLSTM (Schuster & Paliwal, 1997) or pre-trained BERT encoder (Devlin et al., 2019). $Z_i$ ($i \in [1, m]$) represents features such as the NER tags and part-of-speech (POS) tagging. The variable $X$ is the representation of a target relation for RE, entity representation for NER, or trigger representation for ED, and $Y$ indicates the output logits for classification.

Let $\mathcal{E} = \{S, X, Z_1, ..., Z_m\}$ denotes the parents of $Y$. The direct causal effects towards $Y$ including $X \to Y$, $S \to Y$, $Z_1 \to Y$, ...., $Z_m \to Y$ are linear transformations. For each edge $i \to Y$, its transformation is denoted as $\mathbf{W}_{iY} \in \mathbb{R}^{c \times d}$, where $i \in \mathcal{E}$ and $c$ is the number of classes. We let $\mathbf{H}_i \in \mathbb{R}^{d \times h}$ denote $h^1$ representations with $d$ dimensions for node $i \in \mathcal{E}$. Then, the prediction can be obtained by summation $Y_x = \sum_{i \in \mathcal{E}} \mathbf{W}_{iY} \mathbf{H}_i$ or gated mechanism $Y_x = \mathbf{W}_g \mathbf{H}_X \odot \sigma(\sum_{i \in \mathcal{E}} \mathbf{W}_{iY} \mathbf{H}_i)$, where $\odot$ refers to element-wise product, $\mathbf{W}_g \in \mathbb{R}^{c \times d}$ is the linear transformation, and $\sigma(\cdot)$ indicates the sigmoid function. To avoid any single edge, such as $S \to Y$, dominating the generation

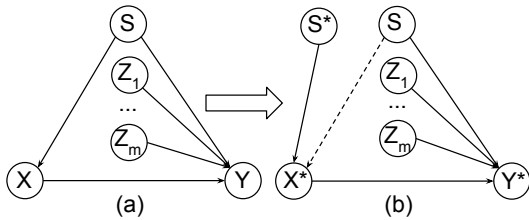

Figure 2: (a) a unified structured causal models for IE tasks. (b) interventions on $X$.

of the logits $Y_x$, we add a cross-entropy loss $\mathcal{L}_{iY}, i \in \mathcal{E}$ for each branch, where $i$ indicates the parent of the node $Y$. Let $\mathcal{L}_Y$ denote the loss for $Y_x$, the total loss $\mathcal{L}$ can be computed by:

$$\mathcal{L} = \mathcal{L}_Y + \sum_{i \in \mathcal{E}} \mathcal{L}_{iY}\tag{2}$$

Note that the proposed SCM is encoder neutral. The SCM can be equipped with various encoders, such as BiLSTM, BERT and Roberta (Liu et al., 2019a). For simplicity, we omit exogenous variables $U$ from the graph as its only useful for the derivations in the following sections.

**Fusing Syntax Structures Into SCMs:** So far we have built basic SCMs for IE tasks. On the edge $S \to X$, we adopt different neural networks architectures for RE, NER and ED. For RE, we use dependency trees to aggregate long-range relations with graph convolution networks (GCN) (Kipf & Welling, 2017). Assume the length of the sentence is $h$. For the GCN, we generate a matrix $\mathbf{A} \in \mathbb{R}^{h \times h}$ from a dependency tree. The convolution computation for the node $i$ at the $l$-th layer takes the representation $\mathbf{x}_i^{l-1}$ from previous layer as input and outputs the updated representations $\mathbf{x}_i^l$. The formulation is given as:

$$\mathbf{x}_i^l = \sigma(\sum_{j=1}^{l} \mathbf{A}_{ij} \mathbf{W}^l \mathbf{x}_i^{l-1} + \mathbf{b}^l), i \in [1, h]\tag{3}$$

where $\mathbf{W}^l$ and $\mathbf{b}^l$ are the weight matrix and bias vector of the $l$-th layer respectively, and $\sigma(\cdot)$ is the sigmoid function. Here $\mathbf{x}^0$ takes value from $\mathbf{H}_S$ and $\mathbf{H}_X$ takes value from the output of the last GCN layer $\mathbf{x}^{l_{max}}$. For NER and ED, we adopt the dependency-guided concatenation approach (Jie & Lu, 2019). Given a dependency edge $(t_h, t_i, r)$ with $t_h$ as a head (parent), $t_i$ as a dependent (child) and $r$ is the dependency relation between them, the representations of the dependent (assume at the

---

[1]$h$ is the sequence length for NER and ED, and $h = 1$ for relation extraction.

$i$-th position of a sentence) can be denoted as:

$$\mathbf{x}_i = [\mathbf{H}_S^{(i)}; \mathbf{H}_S^{(h)}; \mathbf{v}_r], t_h = parent(t_i)$$
$$\mathbf{H}_X = LSTM(\mathbf{x}) \tag{4}$$

where $\mathbf{H}_S^{(i)}$ and $\mathbf{H}_S^{(h)}$ are the word representations of the word $t_i$ and its parent $t_h$, $\mathbf{v}_r$ denotes the learnable embedding of dependency relation $r$.

### 3.2 STEP 2 AND 3: INFERENCE AND COUNTERFACTUAL GENERATION

We have trained our SCMs in the first step. The second step performs inference with the SCMs, and the third step generates dependency-based counterfactuals to better measure the main effect.

**Interventions:** For the SCM $\mathcal{G}$, an intervention indicates an operation that modifies a subset of variables $\mathbf{V} \subseteq \mathbb{V}$ to new values where each variable $V_i \in \mathbf{V}$ is generated by a new structural mechanism $\hat{f}_i(\hat{\mathbf{PA}}_i, U_i)$ that is independent from the original $f_i(\mathbf{PA}_i, U_i)$. Thus, the causal dependency between $V_i$ and its parents $\{\mathbf{PA_i}, U_i\}$ is cutoff. Mathematically, such intervention for one variable $X \in \mathbb{V}$ can be expressed by $do$-notation $do(X = x^*)$ and where $x^*$ is the given value.

**Counterfactuals:** Unlike interventions, the concept of counterfactual reflects an imaginary scenario for "what would the outcome be had the variable(s) been different". Recall from Section 3.1 the definition of SCM and the set of environmental variables $U$ which uniquely determines the variables in the system (Pearl, 2009). Let $Y \in \mathbb{V}$ denote the outcome variable, and let $X \in \mathbb{V}\backslash\{Y\}$ denote the variable of study. The counterfactual for setting $X = x^*$ is formally estimated as:

$$Y_{x^*}(u) = Y_{\mathcal{G}_{x^*}}(u) \tag{5}$$

where $\mathcal{G}_{x^*}$ means assigning $X = x^*$ for all equations in the SCM $\mathcal{G}$. In our CFIE setting, we aim to estimate the counterfactual for the model prediction at instance level. For the proposed SCM shown in Figure 1, the counterfactual $Y_{x^*}$ for our prediction $Y$ is practically computed as follows:

$$Y_{x^*} = Y_{\mathcal{G}_{x^*}}(u) = f_Y(do(X = x^*), S = s, Z = z)$$
$$= \sum_{i \in \mathcal{E}\backslash\{X\}} \mathbf{W}_{iY}\mathbf{H}_i + \mathbf{W}_{XY}\mathbf{H}_{x^*} \tag{6}$$

where $f_Y$ is the function that computes $Y$ and we only replace the original feature representation $\mathbf{H}_X$ with $\mathbf{H}_{x^*}$. No actual value is needed for $u$. See Appendix A.1.1 for derivation.

**Dependency-based Counterfactuals Generation:** There are many other language structures such as constituent tree, abstract meaning representation (Flanigan et al., 2014) and semantic role labeling (Björkelund et al., 2009). We choose the dependency structure in our case as it is able to capture rich relational information as well as complex long-distance interactions that have been proven effective on IE tasks. Counterfactuals lead us to think about: "what are the key clues that determine the relations of two entities for RE, and a certain span of a sentence to be an entity or an event trigger for NER and ED task respectively?". To generate the counterfactual representations for the RE task, we mask the tokens along the shortest path between the two entities of a relation in a dependency tree to form a new sequence. Then this masked sequence is fed to a BiLSTM or BERT encoder to output new contextualized representations $S^*$. For the NER and ED task, we mask entities, or the tokens in the scope of 1 hop on the dependency tree to generate $S^*$. Then we feed $S^*$ to the function $S \rightarrow X$ to get $X^*$. The operation on NER also aligns a recent finding (Zeng et al., 2020) that the entity itself is more important than context for entity classification. By doing so, the key clues have been wiped off in the generated counterfactuals representations $X^*$, which can be used to strengthen the main effect while reduce spurious correlations and the side effect.

### 3.3 STEP 4 AND 5: CAUSAL EFFECT ESTIMATION

We estimate the causal effect in the fourth step and make use of the couterfactuals representation for a more robust prediction in the fifth step. Inspired by Total Direct Effect (TDE) used in (Tang et al., 2020b), we can compare the original outcome $Y_x$ and its counterfactual $Y_{x^*}$ to estimate the effect of RE so that the side effect can be eliminated (see Appendix A.1.2 for derivation):

$$TDE = Y_x - Y_{x^*} \tag{7}$$

As both context and entity (or trigger) play important roles for the classification in the NER and ED tasks, we propose a novel approach to alleviate the spurious correlations caused by side effects, while strengthening the main effect at the same time. The interventional causal effect of the $i$-th entity in a sequence can be described as:

$$Effect = Y_{x_i} - Y_{x_i^*} + \alpha \mathbf{W}_{XY} \mathbf{x}_i^* \tag{8}$$

where $\alpha$ is the hyperparameter that balances the importance of context and entity (or trigger) for the NER and ED task. The first part $Y_{x_i} - Y_{x_i^*}$ indicates the main effect, which reflects more about the debiased context, while the second part $\mathbf{W}_{XY} \mathbf{x}_i^*$ reflects more about the entity (or trigger) itself. Combining them yields more robust prediction by better distinguishing the main and side effect. As shown in Figure 1, the sentence "The program was killed" produces biased high score for event "Life:Die" in $Y_x$ and results in wrong prediction due to the word "killed". By computing the counterfactual $Y_{x^*}$ with "program" masked, the score for "Life:Die" remains high but the score for "SW:Quit" drops dramatically. This difference $Y_{x_i} - Y_{x_i^*}$ leads us to correct prediction and knowing the important role of the word "program". Such a design differs from that of the previous work used in vision community (Tang et al., 2020a) by providing more flexible adjustment and effect estimation. We will show that our approach is more suitable for long-tailed IE tasks.

## 4 EXPERIMENTS

### 4.1 DATASETS AND SETTINGS

The five datasets used in our experiments include OntoNotes5.0 (Pradhan et al., 2013) and ATIS (Tur et al., 2010) for the NER task, ACE2005 (Doddington et al., 2004) and MAVEN (Wang et al., 2020b) for the ED task, and NYT24 (Gardent et al., 2017) for the RE task. For all the five datasets, we categorize the classes into three splits based on the number of training instances per class. The model parameters are finetuned on the development sets. For RE, we leverage Stochastic Gradient Descent (SGD) optimizer with a 0.3 learning rate and 0.9 weight decay rate. For NER and ED, we utilize Adam optimizer with an initial learning rate of 0.001. The hidden size of the BiLSTM and GCNs are set as 300, and the number of layers of GCNs is configured as 3. 300-dimensional GloVe (Pennington et al., 2014) is used to initialize the word embeddings [2]. We focus more on Mean Recall (MR) (Tang et al., 2020b) and Macro F1 (MF1), two more balanced metrics to measure the performance of long-tailed IE tasks, as MR is able to better reflect the capability in identifying the instance-scare class, and MF1 can better represent the model's ability for each class, while the conventional Micro F1 score highly depends on the data-rich classes and pays less attention to the tail classes. We report the Micro F1 score (F1) for each dataset in the Appendix. We also follow (Liu et al., 2019b) to report the MR and MF1 on three splits in Table 5 in the Appendix.

### 4.2 BASELINES

We categorized the baselines into three groups. 1) **Conventional Models** include BiLSTM (Chiu & Nichols, 2016), BiLSTM+CRF (Ma & Hovy, 2016), C-GCN (Zhang et al., 2017), Dep-Guided LSTM (Jie & Lu, 2019), AGGCN (Guo et al., 2019) and BERT (Devlin et al., 2019). They do not explicitly take the long-tailed issues into consideration. 2) **Re-weighting/Decoupling models** refer to loss re-weighting approaches including Focal Loss (Lin et al., 2017), and two-stage decoupled learning approaches (Kang et al., 2019) that include $\tau$-normalization, classifier retraining (cRT) and learnable weight scaling (LWS). 3) **Causal model** include TDE (Tang et al., 2020b). There are also recent studies based on the deconfounded methodology (Tang et al., 2020a; Yang et al., 2020), which however seem not applicable to be selected as a causal baseline in our case. In our experiments, we reproduced the results for all the baselines as most of the results have not been reported on NLP datasets. We believe some recent strong baselines, which are not mentioned in this paper due to space limitation, may also further benefit our model by integrating them into the edge $S \rightarrow X$.

### 4.3 TASK DEFINITIONS

**Named Entity Recognition:** NER is a sequence labeling task that seeks to locate and classify named entities in unstructured text into pre-defined categories such as person, location, etc. **Event**

---

[2]The statistics of the datasets and detailed hyperparameters are attached in the Appendix

Table 1: Evaluation results on the OntoNotes5.0 dataset and ATIS dataset for the NER task.

| Model | OntoNotes5.0 | | | | ATIS | | | |
| | Few | | Overall | | Few | | Overall | |
| | MR | MF1 | MR | MF1 | MR | MF1 | MR | MF1 |
|---|---|---|---|---|---|---|---|---|
| BiLSTM (Chiu & Nichols, 2016) | 67.5 | **69.9** | 76.4 | 76.8 | 66.2 | 69.0 | 84.2 | 83.1 |
| BiLSTM+CRF (Ma & Hovy, 2016) | 60.7 | 63.6 | 71.6 | 73.5 | 58.1 | 60.4 | 81.0 | 80.2 |
| C-GCN (Zhang et al., 2017) | 68.3 | 69.8 | 77.3 | 76.8 | 63.2 | 65.3 | 82.0 | 80.1 |
| Dep-Guided LSTM (Jie & Lu, 2019) | 61.8 | 69.3 | 74.1 | 75.8 | 60.6 | 65.4 | 84.6 | 84.3 |
| Focal Loss (Lin et al., 2017) | 64.1 | 65.5 | 74.2 | 73.9 | 48.9 | 49.8 | 78.7 | 76.6 |
| cRT (Kang et al., 2019) | 64.1 | 68.5 | 75.0 | 76.1 | 68.1 | 71.7 | 85.7 | 84.8 |
| $\tau$ - Normalization (Kang et al., 2019) | 61.1 | 66.7 | 73.5 | 75.7 | 64.8 | 68.0 | 83.9 | 83.1 |
| LWS (Kang et al., 2019) | 58.7 | 64.9 | 72.1 | 74.7 | 66.2 | 69.1 | 84.3 | 83.2 |
| TDE (Tang et al., 2020b) | 71.9 | 68.8 | 80.4 | 76.7 | 67.5 | 67.1 | 87.1 | 84.5 |
| Ours (Glove) | **76.7** | 68.9 | **83.8** | **77.3** | **71.8** | **73.1** | **88.6** | **87.0** |
| BERT (Devlin et al., 2019) | 77.7 | 76.5 | 84.6 | 82.4 | 52.3 | 56.1 | 82.9 | 81.7 |
| Roberta (Liu et al., 2019a) | 78.7 | **79.5** | 86.5 | **85.7** | **60.2** | 61.1 | 84.3 | 82.3 |
| BERT+GCN(Wadden et al., 2019) | 80.2 | 77.7 | 85.6 | 82.6 | 53.3 | 56.6 | 83.0 | 81.3 |
| Ours (BERT) | **80.6** | 79.1 | **86.7** | 84.1 | 58.8 | **62.2** | **85.5** | **82.6** |

Table 2: Evaluation results on the ACE2005 dataset and MAVEN dataset for the event detection.

| Model | ACE2005 | | | | MAVEN | | | |
| | Few | | Overall | | Few | | Overall | |
| | MR | MF1 | MR | MF1 | MR | MF1 | MR | MF1 |
|---|---|---|---|---|---|---|---|---|
| BiLSTM (Chiu & Nichols, 2016) | 34.2 | 35.6 | 52.3 | 54.8 | 36.5 | 40.7 | 67.1 | 69.5 |
| BiLSTM+CRF (Ma & Hovy, 2016) | 41.4 | 45.1 | 51.8 | 54.1 | 43.4 | 46.8 | 69.6 | 71.1 |
| C-GCN (Zhang et al., 2017) | 41.4 | 44.1 | 52.0 | 56.1 | 49.7 | 51.7 | 73.1 | 73.0 |
| Dep-Guided LSTM (Jie & Lu, 2019) | 42.8 | 41.7 | 52.4 | 55.8 | 44.7 | 45.4 | 67.8 | 69.3 |
| Focal Loss (Lin et al., 2017) | 38.6 | 42.9 | 52.6 | 58.5 | 45.4 | 51.5 | 70.3 | 73.8 |
| cRT (Kang et al., 2019) | 44.8 | 47.4 | 57.6 | 58.9 | 49.7 | 55.4 | 71.0 | 74.6 |
| $\tau$ - Normalization (Kang et al., 2019) | 34.3 | 35.6 | 53.3 | 52.5 | 21.1 | 26.7 | 51.0 | 58.4 |
| LWS (Kang et al., 2019) | 34.3 | 35.6 | 58.2 | 56.9 | 33.3 | 38.7 | 65.9 | 68.7 |
| TDE (Tang et al., 2020b) | 34.3 | 33.9 | 58.5 | 56.5 | 39.8 | 36.2 | 71.9 | 67.4 |
| Our (Glove) | **47.1** | **49.7** | **63.5** | **60.2** | **60.4** | **57.4** | **79.8** | **76.0** |
| BERT(Devlin et al., 2019) | 47.6 | 48.9 | 66.5 | 65.1 | 38.1 | 39.3 | 73.5 | 72.3 |
| Roberta (Liu et al., 2019a) | 47.6 | 47.7 | 70.2 | 68.6 | **43.1** | 43.8 | **75.0** | **73.6** |
| BERT+GCN (Wadden et al., 2019) | 45.2 | 47.5 | 73.1 | 71.3 | 40.9 | 42.0 | 74.1 | 72.8 |
| Ours (BERT) | **61.9** | **63.2** | **74.9** | **74.4** | **43.1** | **45.0** | 71.3 | 71.7 |

**Detection:** ED aims to detect the occurrences of predefined events and categorize them as triggers from unstructured text. Event trigger is defined as the words or phase that most clearly expresses an event occurrence. Taking the sentence "a cameraman died in the Palestine Hotel" as an example, the word "died" is considered as the trigger with a "Death" event. **Relation Extraction:** The goal of RE is to identify semantic relationships from text, given two or more entities. For example, "Paris is in France" states a "is in" relationship between two entities Paris to France. Their relation can be denoted by the triples (Paris, is in, France).

### 4.4 RESULTS

**Named Entity Recognition:** Table 1 shows the comparison results on both OntoNotes5.0 and ATIS datasets. Our models outperform the two classical models `BiLSTM` and `BiLSTM+CRF` under most settings, especially on the Few setting, e.g achieving 10.2 points higher Mean Recall (MR) against `BiLSTM` on OntoNotes5.0, and 12.7 points higher Mean F1 (MF1) against `BiLSTM+CRF` on ATIS. The results indicate the superiority of our proposed model in handling the instance-scarce classes. Comparing with the C-GCN model that makes use of dependency trees for information aggregation, our model also achieves 8.4 higher MR and comparable MF1, indicating the capability of a causal model in improving the long-tailed sequence labeling problem. Comparing with a recent causal baseline TDE, our model consistently perform better in terms of long-tailed scores, the results confirm our hypothesis that making good use of language structure helps a causal model to distinguish main effect from the side effect. Among re-balancing approaches such as Focal Loss, cRT and LWS, $\tau$-Normalization performs best and this aligns with the findings in the previous study (Kang et al., 2019) for long-tailed image classification.

**Event Detection:** Table 2 shows comparison results on both ACE2005 and MAVEN datasets. Overall, our model significantly outperforms the baselines under the Few setting by a large margin, e.g., 12.8 and 15.8 pointers higher in terms of MR and MF1 respectively on ACE2005 dataset, 20.6 and 20.8 points higher in terms of the two metrics on MAVEN dataset. Meanwhile, our model is able to achieve better or comparable results under other settings. The results further confirm the robustness of our model in improving the classifications for tail classes with few training instances available. Our model also performs better than BERT baselines under the Few setting, indicating that the pre-trained BERT models still suffer bias issues on the long-tailed IE tasks.

**Relation Extraction:** As shown in Table 3, we further evaluate `CFIE` for the relation extraction on NYT24 dataset. Our method significantly outperforms all other methods in MF for both tail classes and overall F1. Although cRT achieves relatively high MR, having the lowest MF1 renders it incompetent for this task. The results further confirm our hypothesis that the proposed `CFIE` is able to alleviate spurious correlations caused by

Table 3: Evaluation results on the NYT24 dataset for RE.

| Model | NYT24 | | | |
| | Few | | Overall | |
| | MR | MF1 | MR | MF1 |
| --- | --- | --- | --- | --- |
| C-GCN (Zhang et al., 2017) | 24.0 | 26.7 | 51.2 | 52.6 |
| Focal Loss (Lin et al., 2017) | 52.0 | 48.3 | 62.9 | 61.9 |
| cRT (Kang et al., 2019) | 66.0 | 24.2 | **65.6** | 50.5 |
| $\tau$ - Normalization (Kang et al., 2019) | 40.0 | 40.0 | 53.5 | 54.6 |
| LWS (Kang et al., 2019) | 40.0 | 40.0 | 53.5 | 54.6 |
| TDE (Tang et al., 2020b) | 60.0 | 57.1 | 61.0 | 60.2 |
| Ours (Glove) | **68.0** | **68.6** | 65.3 | **63.6** |

imbalanced dataset by learning to distinguish the main effect from the side effect. We also observe that `CFIE` outperforms the previously proposed TDE by a large margin for the both Few and Overall settings, i.e., 11.5 points and 3.4 points improvement in terms of MF1. This further proves our hypothesis that properly exploring language structure on causal models will boost the performance of IE tasks on imbalanced datasets.

## 4.5 DISCUSSIONS

**What are the most important factors for NER?** We have hypothesised that the factors, such as 2-hop and 1-hop context on the dependency tree, the entity itself, and POS feature, may hold the potential to be the key clues for the NER predictions. To evaluate the impact of these factors, we first generate new sequences by masking or mit-

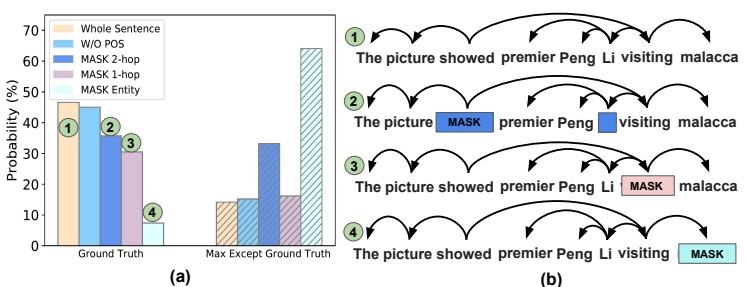

Figure 3: (a) prediction distributions for various factors. (b) masking operations based on a syntax tree.

igating these factors. Then we feed the generated sequences to the proposed SCM to obtain the predictions. Figure 3 shows a qualitative example for predicting the NER tag for the entity "malacca". Specifically, Figure 3 (a) visualizes the variances of the predictions, where the histograms in the left refer to prediction probabilities for the ground truth class, while the histograms in the right are the max predictions except the results of ground truth class. Figure 3(b) illustrates how we mask the context based on a dependency tree. It shows that masking the entity, i.e., "malacca", will lead to the most significant performance drop, indicating that entity itself plays a key role for the NER classification. This also inspires us to design step 5 in our framework. More analyses about ED and RE are given in the Appendix A.4.1 and A.4.2.

**Does the syntax structure matter?** To answer this question, we design three baselines including: 1) `Causal Models w/o Syntax` that doesn't employ dependency trees during the training stage, and only uses it for generating counterfactuals, 2) `Counterfactuals w/o Syntax` that employs dependency structures for training but utilizes a null input as the intervention during the inference state. We refer such a setting from the previous study (Tang et al., 2020a), and 3) `No Syntax` that is the same to the previous work TDE (Tang et al., 2020b) which don't involve depen-

dency structures in both training and inference stages. As shown in Figure 4, our model outperforms the first two baselines on the ACE2005 dataset under both Few and All settings, demonstrating the effectiveness of dependency structure in improving the causal models for long-tailed IE.

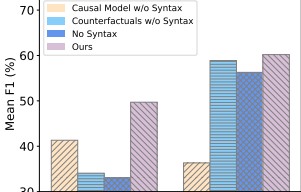 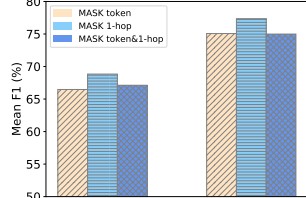 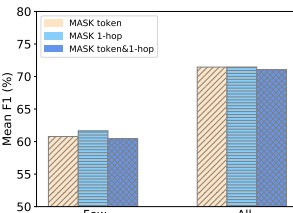

Figure 4: Syntax Contribution.   Figure 5: Prune with DGLSTM.   Figure 6: Prune with C-GCN.

**How can we make good use of dependency structure?** To answer this question, we present three tree pruning mechanisms under two graph aggregation settings, i.e., `Prune with DGLSTM` and `Prune with C-GCN` as described in Equation 3 and Equation 4. The three pruning strategies include 1) `CFIE Mask 1-hop` which masks the tokens that directly connect to the targeting token in a dependency tree, 2) `CFIE Mask token` which directly masks the targeting token, 3) `CFIE Mask token&1-hop` which masks both the targeting token and its 1-hop neighbours in the dependency tree. Figure 5 and Figure 6 depict the results on OntoNotes5.0 dataset. We observe that masking 1-hop neighbours in the dependency tree achieves the best performance among three strategies, indicating that an entity itself is more important in NER sequence labeling. By comparing the two graph aggregation method, we draw a conclusion that `Prune with DGLSTM` can make better use of dependency structures.

**How about the performance under various interventions and SCMs?** We study this question on ACE2005 dataset for ED task. We design three interventional methods including 1) `Intervene X & NER`, 2) `Intervene X & POS`, 3) `Intervene X & NER & POS`. Figure 7 shows that introducing interventions solely on $X$ is able to achieve the best performance under both Few and All settings. We also introduce three variants of our proposed SCMs : 1) `SCM w/o NER`, 2) `SCM w/o POS`, 3) `SCM w/o NER and POS`. Figure 8 shows that mitigating the NER node will significantly decrease the ED performance, especially over the Few setting. The results prove the superiority of our proposed SCMs that explicitly involve linguistic features to calculate main effect. More analyses for NER task are given in Appendix A.4.4.

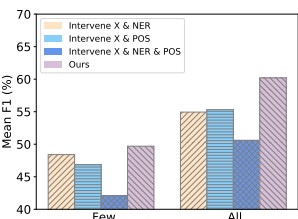 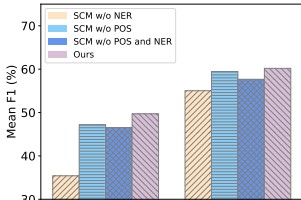 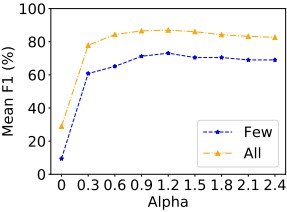

Figure 7: Various interventions.      Figure 8: Various SCMs.        Figure 9: Various $\alpha$.

**How the hyper-parameter $\alpha$ impacts the performance?** To evaluate the impact of $\alpha$ on the performance, we tuned the parameter on four datasets including OntoNotes5.0, ATIS, ACE2005, and MAVEN. As shown in Figure 9, when increasing $\alpha$ from 0 to 2.4 on ATIS dataset, the F1 scores increase at first dramatically then decrease slowly. The F1 scores reach the peak when $\alpha$ is set to 1.2. As the value of $\alpha$ represents the importance of entity for classifications, we therefore draw a conclusion that, for NER task , an entity plays a relatively more important role than the context. We also demonstrate the necessity of step 5 in our framework, since the performance is poor when $\alpha$ is set to 0. Experimental results on the other three datasets are given in the Appendix A.4.3.

## 5   CONCLUSION

In this paper, we present `CFIE`, a novel approach to tackling the long-tailed information extraction issues via counterfactual analysis in causal inference. Experimental results on five datasets across three IE tasks show the effectiveness of our approach. The future research directions include applying the proposed framework to more challenging long-tailed document-level IE tasks.

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

## A  APPENDIX

### A.1  DERIVATIONS

#### A.1.1  COUNTERFACTUALS

Recall that the formal computation for counterfactual is defined as:

$$Y_{x^*}(u) = Y_{M_{x^*}}(u)$$

where $M_{x^*}$ means assigning $X = x^*$ for all equations in the SCM. The crucial step in the derivation is to understand the goal of the exogenous variable $U$, by which the variables in the causal graph are uniquely determined. To compute the counterfactual of a prediction regarding variable $X$, we have to keep all other variables under the same setting as the original prediction. Consider an intuitive example that a boy got an A for the subject because he studied hard. To estimate the counterfactual "what score would he get if he did not study hard", we should maintain all other factors like the difficulty of the subject and the skills of the teacher and so on at the original level to simulate the hypothetical scenario that the boy travelled back in time and behaved differently. Thus, setting $U = u$ where $u$ is the environment (e.g. year of admission, faculty) for the original prediction is to ensure the consistency in estimating the value of all other variables, which is mathematically:

$$V_i = f_i(\mathbf{PA}_i, U = u), \forall V_i \in \mathbb{V}$$

except for the variable of interest $X$ along with its descendants (e.g. commendation from the teacher) due to the intervention $do(X = x^*)$. Thus, for our SCM, as long as we can ensure the value of variables $(S, Z)$ that are not descendants of $X$ follow the original situation, the exogenous variable $u$ is only for notational purpose and no longer needed in computing the counterfactuals. Besides, for all variables, only the descendants of $X$ should be re-calculated. We now present the mathematical derivation for the counterfactual $Y_{x^*}$ in our SCM:

$$
\begin{aligned}
Y_{x^*} &= Y_{M_{x^*}}(u) \\
&= Y(do(X = x^*), U = u) \\
&= f_Y(do(X = x^*), S = s, Z = z) \\
&= f_Y(x^*, s, z) \\
&= \sum_{i \in \mathcal{E} \setminus \{X\}} \mathbf{W}_{iY}\mathbf{H}_i + \mathbf{W}_{XY}\mathbf{H}_{x^*}
\end{aligned}
$$

In short, to compute the counterfactual $Y_{x^*}$, we simply need to

1. Assign a new value $x^*$ to the variable of interest $X$.
2. Cut off the dependency between $X$ and its parents in SCM.
3. Recompute all values.

### A.1.2 TOTAL DIRECT EFFECT

In an SCM, let $M$ be the mediator variables such that path $X \rightarrow Z \rightarrow Y$ exists. The formal definition of Total Direct Effect (TDE) is:

$$TDE = Y_x(u) - Y_{x^*,m}(u)$$

where $m$ are the original values of the mediator variables. Thus, additional intervention $do(M = m)$ is required to compute $TDE$. Fortunately, our SCM shown in Figure 1 does not have mediators for $X$ and the computation is reduced to:

$$TDE = Y_x(u) - Y_{x^*}(u)$$
$$= Y_x - Y_{x^*}$$

One may question that why $X$ imposes no effect on $Z$ including POS and NER tags for relation extraction. This is because POS and NER tags are provided in the dataset and we are not using them for joint training. Thus, there is no direct dependency between contextual representation and the representation for the tags.

### A.2 DATASET STATISTICS

We give the statistics of five datasets as follows in Table 4. We follow the (Liu et al., 2019b) to split the training set as Few-shot(Few), Medium-shot(Medium) and Many-Shot(Many). We split the dataset based on the distribution of class types and numbers. Details are given in Table 5.

Table 4: Data Statistics

| Dataset | Task | Train | Dev | Test | Class Types |
|---------|------|-------|-----|------|-------------|
| OntoNotes5.0 | Named Entity Recognition | 59924 | 8528 | 8262 | 18 |
| ATIS | Named Entity Recognition | 4479 | 498 | 893 | 79 |
| ACE2005 | Event Detection | 19216 | 901 | 676 | 33 |
| MAVEN | Event Detection | 32431 | 8042 | 9400 | 168 |
| NYT24 | Relation Extraction | 50577 | 5619 | 5000 | 24 |

Table 5: Number of instances per class for dataset splitting

| Dataset | Task | Few | Medium | Many |
|---------|------|-----|--------|------|
| OntoNotes5.0 | Named Entity Recognition | $\leq 4000$ | $4000 \sim 10000$ | $\geq 10000$ |
| ATIS | Named Entity Recognition | $\leq 30$ | $30 \sim 100$ | $\geq 100$ |
| ACE2005 | Event Detection | $\leq 30$ | $30 \sim 150$ | $\geq 150$ |
| MAVEN | Event Detection | $\leq 100$ | $100 \sim 1000$ | $\geq 1000$ |
| NYT24 | Relation Extraction | $\leq 100$ | $100 \sim 200$ | $\geq 200$ |

### A.3 EXPERIMENT SETTINGS

We use spaCy[3] to generate the dependency tree, NER as well as POS tagging for a input sentence. The hyperparameters that we used on three tasks are listed as follows in Table 6, Table 7, and Table 8. We show the parameters in different tables as the setting varies for each task.

### A.4 MORE DISCUSSIONS

We add more discussions here based on Section 4.5.

---

[3]https://spacy.io/

### A.4.1 What are the most important factors for the ED task?

To answer this question, we conduct experiments on ACE2005. We have hypothesised that the factors, such as 2-hop and 1-hop context on the dependency tree, the entity itself, POS feature, and NER feature may hold the potential to be the key clues for the ED predictions. The design of

Table 6: Detailed Hyper-parameters for the NER task.

| | |
|---|---|
| Batch size | 64 |
| Learning rate | 0.001 |
| Decay rate | 0.90 |
| Gradient clipping | 5 |
| Optimizer | Adam |
| Word embedding dimension | 300 |
| Input dropout | 0.3 |
| RNN Hidden size | 200 |
| RNN Layer size | 1 |
| RNN dropout | 0.3 |
| POS dim | 30 |
| Deprel dim | 30 |
| $\alpha$(OntoNotes5.0) | 0.9 |
| $\alpha$(ATIS) | 1.2 |

Table 7: Detailed Hyper-parameters for the ED task.

| | |
|---|---|
| Batch size | 32 |
| Learning rate | 0.001 |
| Decay rate | 0.90 |
| Gradient clipping | 5 |
| Optimizer | Adam |
| Word embedding dimension | 300 |
| Input dropout | 0.3 |
| RNN Hidden size | 200 |
| RNN Layer size | 1 |
| RNN dropout | 0.3 |
| POS dim | 30 |
| NER dim | 30 |
| Deprel dim | 30 |
| $\alpha$ | 1.5 |

Table 8: Detailed Hyper-parameters for the RE task.

| | |
|---|---|
| Batch size | 50 |
| Learning rate | 0.03 |
| Decay rate | 0.90 |
| Gradient clipping | 5 |
| Optimizer | SGD |
| Word embedding dimension | 300 |
| Input dropout | 0.5 |
| RNN Hidden size | 300 |
| RNN Layer size | 2 |
| RNN dropout | 0.2 |
| Block Number | 2 |
| GCN first sub-layers | 2 |
| GCN second sub-layers | 4 |
| GCN dropout | 0.3 |
| Pooling L2 | 0.002 |

our experiments here are similar as that of NER task described in Section 4.5. Figure 10 shows a qualitative example for predicting the event type for the word "shot". Specifically, Figure 10 (a) visualizes the variances of the predictions, where the histograms in the left refer to prediction probabilities for the ground truth class, while the histograms in the right are the max predictions except the results of ground truth class. Figure 10(b) illustrates how we mask the context based on a dependency tree. We obtain the same conclusion that masking the word itself, i.e., "shot", will lead to the most significant performance drop, indicating that entity itself serves as a key for the ED classification. Also we can see that 1-hop neighbors in the dependency tree plays the second important roles. When 1-hop neighbors are masked, the lead in the probability of the ground truth class is reduced relative to the probability of the error class, which indicates the decline of the model's classification ability.

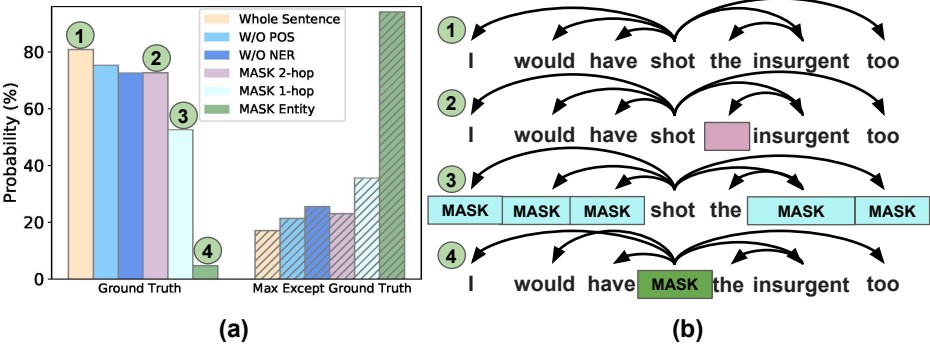

Figure 10: (a) prediction distributions for various factors. (b) masking operations based on a syntax tree.

### A.4.2 WHAT ARE THE MOST IMPORTANT FACTORS FOR THE RE TASK?

To answer this question, we conduct experiments on NYT24 dataset. We have hypothesised that the factors, such as context on the shortest path between the targets, the contextualized word representations, POS feature, and NER feature may hold the potential to be the key clues for the RE predictions. The design of our experiments here are similar as that of NER task described in Section 4.5. Figure 11 shows a qualitative example for predicting the relation type for the targets "Italy" and "Modena". Specifically, Figure 11 (a) visualizes the variances of the predictions, where the histograms in the left refer to prediction probabilities for the ground truth class, while the histograms in the right are the max predictions except the results of ground truth class. Figure 11(b) illustrates how we mask the context based on a dependency tree. When we mask the context on the shortest path, we can see that the probability on the ground truth class drops significantly and the model makes a wrong prediction, which indicates the importance of context on the shortest path between subject and object in RE task.

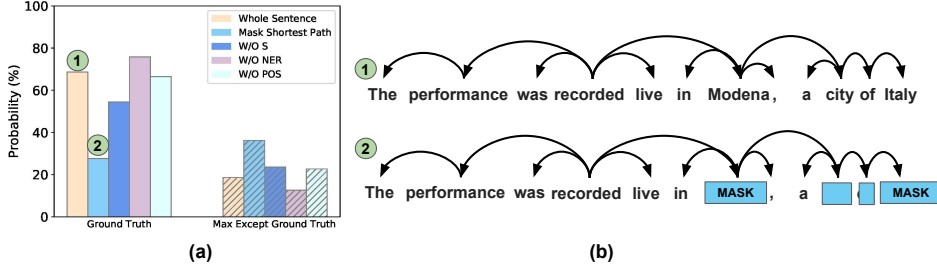

Figure 11: (a) prediction distributions for various factors. (b) masking operations based on a syntax tree.

### A.4.3 HOW THE HYPER-PARAMETER $\alpha$ IMPACTS THE PERFORMANCE?

Here we show the performance on OntoNotes5.0, ACE2005, and MAVEN datasets regarding various values of $\alpha$. As shown in Figure 12, we observe that the trends are similar on different datasets. The optimal values are 0.9, 1.5, 1.5 respectively on OntoNotes5.0, ACE2005, and MAVEN dataset.

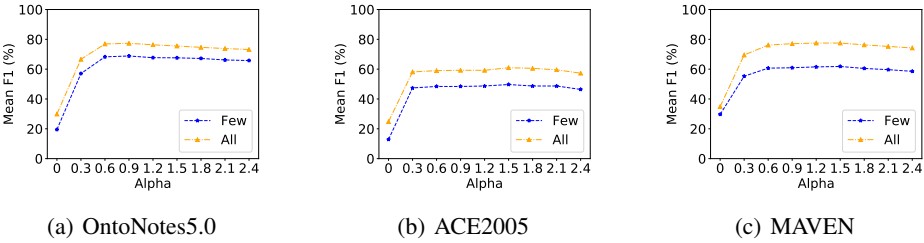

(a) OntoNotes5.0  (b) ACE2005  (c) MAVEN

Figure 12: Performance with different values of $\alpha$

### A.4.4 Exploring different interventions and SCMs for NER task.

We conduct experiments for NER task on OntoNotes5.0 dataset regarding different intervention methods and SCMs. The design and conclusions are similar to those of the ED task described in Section 4.5. The results are shown in Table 13. To be specific, only intervening X achieves the best performance, indicating that our method is capable of capturing the most significant effect. Furthermore, our design of including POS tag in the causal graph can incorporate the useful information while eliminating the bias in POS tags.

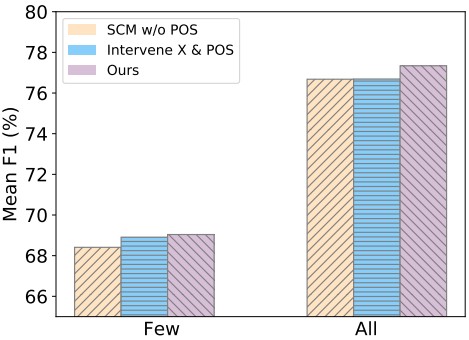

Figure 13: Performance with various interventions and SCMs for NER task.

### A.5 Measuring causal effects of various factors

We measure the causal effects of different factors for RE task. Here we define a set of factors $\mathbb{F} = \{X, S, NER, POS, TAGS, Context, DepEdges\}$, where $S, X, NER, POS$ are variables defined in our SCM, $TAGS$ includes both NER tag and POS tag, $Context$ denotes tokens along the shortest path between subject and object in RE task, and $DepEdges$ denotes the dependency edges connected to either subject or object. We calculate the causal effect by Equation 7, where $x^*$ is generated by masking each factor in $\mathbb{F}$. Instead of measuring the effect on a specific instance, we calculate the average effect on the ground truth class over all samples in NYT24 dataset. A larger value indicates a more significant causal effect from the specific factor to the ground truth label. From Table 9 we can observe that $X$ and $Context$ have the largest effect to the ground truth, which is captured in our model. Also we can conclude that masking tokens in the dependency tree is better choice compared with masking dependency relations.

Table 9: Average causal effect from various factors to the ground truth label.

| Factor | $X$ | $S$ | $NER$ | $POS$ | $TAGS$ | $Context$ | $DepEdges$ |
|---|---|---|---|---|---|---|---|
| Avg. Effect | 0.656 | 0.502 | 0.108 | 0.027 | 0.074 | 0.633 | 0.061 |

## A.6   MORE DETAILED EXPERIMENTAL RESULTS

For the NER and ED tasks, we report more detailed comparisons on the Ontonotes5.0, ATIS, ACE2005, and MAVEN datasets in Table 10, Table 11, Table 12, and Table 13 respectively. We also report the detailed results for RE on the NYT24 dataset in Table 14.

Table 10: Evaluation results on the OntoNotes5.0 dataset for the named entity recognition.

| Model | Few | | Medium | | Many | | Overall | | |
|---|---|---|---|---|---|---|---|---|---|
| | MR | MF1 | MR | MF1 | MR | MF1 | MR | MF1 | Micro F1 |
| BiLSTM (Chiu & Nichols, 2016) | 67.5 | **69.9** | 72.6 | 75.3 | 88.1 | 85.4 | 76.4 | 76.8 | 83.7 |
| BiLSTM+CRF (Ma & Hovy, 2016) | 60.7 | 63.6 | 65.3 | 69.1 | 86.9 | 86.9 | 71.6 | 73.5 | **85.5** |
| C-GCN (Zhang et al., 2017) | 68.3 | 69.8 | 69.1 | 72.9 | 90.9 | 86.6 | 77.3 | 76.8 | **85.5** |
| Dep-Guided LSTM (Jie & Lu, 2019) | 61.8 | 69.3 | 70.2 | 73.7 | 89.8 | 84.3 | 74.1 | 75.8 | 84.0 |
| Focal Loss (Lin et al., 2017) | 64.1 | 65.5 | 69.9 | 71.2 | 87.7 | 84.7 | 74.2 | 73.9 | 85.2 |
| cRT (Kang et al., 2019) | 64.1 | 68.5 | 73.9 | 75.3 | 88.0 | 85.2 | 75.0 | 76.1 | 83.6 |
| $\tau$ - Normalization (Kang et al., 2019) | 61.1 | 66.7 | 72.8 | **76.4** | 88.0 | 85.7 | 73.5 | 75.7 | 84.9 |
| LWS (Kang et al., 2019) | 58.7 | 64.9 | 71.6 | 76.1 | 87.6 | 85.2 | 72.1 | 74.7 | 84.9 |
| TDE (Tang et al., 2020b) | 71.9 | 68.8 | 77.9 | 74.8 | 91.2 | 86.7 | 80.4 | 76.7 | 83.6 |
| Ours (Glove) | **76.7** | 68.9 | **83.6** | 76.2 | **92.0** | **87.6** | **83.8** | **77.3** | 85.4 |
| BERT (Devlin et al., 2019) | 77.7 | 76.5 | 81.4 | 78.6 | 94.0 | 90.7 | 84.6 | 82.4 | 88.7 |
| Roberta (Liu et al., 2019a) | 78.7 | **79.5** | 84.8 | **85.1** | **96.3** | **93.1** | 86.5 | **85.7** | **90.7** |
| BERT+GCN (Wadden et al., 2019) | 80.2 | 77.7 | 81.1 | 77.3 | 93.9 | 90.4 | 85.6 | 82.6 | 88.6 |
| Ours (BERT) | **80.6** | 79.1 | **85.1** | 80.4 | 94.5 | 91.4 | **86.7** | 84.1 | 88.9 |

Table 11: Evaluation results on the ATIS dataset for the named entity recognition.

| Model | Few | | Medium | | Many | | Overall | | |
|---|---|---|---|---|---|---|---|---|---|
| | MR | MF1 | MR | MF1 | MR | MF1 | MR | MF1 | Micro F1 |
| BiLSTM | 66.2 | 69.0 | 89.8 | 85.9 | 93.0 | 92.2 | 84.2 | 83.1 | 92.0 |
| BiLSTM+CRF | 58.1 | 60.4 | 87.4 | 83.5 | 93.0 | **93.5** | 81.0 | 80.2 | 93.2 |
| CGGCN (Zhang et al., 2017) | 63.2 | 65.3 | 87.3 | 83.2 | 91.8 | 89.3 | 82.0 | 80.1 | 91.8 |
| Dep-Guided LSTM (Jie & Lu, 2019) | 60.6 | 65.4 | 94.1 | 90.4 | 93.2 | 92.9 | 84.6 | 84.3 | **93.8** |
| Focal Loss | 48.9 | 49.8 | 89.3 | 84.6 | 91.1 | 89.8 | 78.7 | 76.6 | 90.0 |
| cRT | 68.1 | 71.7 | 92.3 | 88.0 | 92.8 | 92.2 | 85.7 | 84.8 | 92.8 |
| $\tau$ - Normalization | 64.8 | 68.0 | 89.9 | 86.2 | 93.0 | 92.5 | 83.9 | 83.1 | 93.2 |
| LWS | 66.2 | 69.1 | 89.9 | 85.9 | 93.0 | 92.2 | 84.3 | 83.2 | 93.0 |
| TDE (Tang et al., 2020b) | 67.5 | 67.1 | 95.3 | 89.4 | 93.7 | 93.5 | 87.1 | 84.5 | 92.4 |
| Ours (Glove) | **71.8** | **73.1** | 95.6 | 91.4 | 94.3 | 93.5 | **88.6** | **87.0** | 92.8 |
| BERT(Devlin et al., 2019) | 52.3 | 56.1 | 96.0 | **90.6** | 92.9 | **92.7** | 82.9 | 81.7 | 94.5 |
| Roberta(Liu et al., 2019a) | **60.2** | 61.1 | 93.8 | 89.1 | 93.1 | 92.2 | 84.3 | 82.3 | 94.3 |
| BERT+GCN (Wadden et al., 2019) | 53.3 | 56.6 | 95.4 | 89.6 | 93.1 | 92.2 | 83.0 | 81.3 | 94.5 |
| Ours (BERT) | 58.8 | **62.2** | 97.2 | 89.1 | **93.6** | 92.4 | **85.5** | 82.6 | **94.6** |

Table 12: Evaluation results on the ACE2005 dataset for the event detection.

| Model | Few | | Medium | | Many | | Overall | | |
|---|---|---|---|---|---|---|---|---|---|
| | MR | MF1 | MR | MF1 | MR | MF1 | MR | MF1 | Micro F1 |
| BiLSTM  (Chiu & Nichols, 2016) | 34.2 | 35.6 | 55.1 | 58.2 | 64.9 | 67.0 | 52.3 | 54.8 | 66.7 |
| BiLSTM+CRF  (Ma & Hovy, 2016) | 41.4 | 45.1 | 49.8 | 52.2 | 70.1 | 70.5 | 51.8 | 54.1 | 68.2 |
| C-GCN (Zhang et al., 2017) | 41.4 | 44.1 | 51.2 | 55.8 | 66.4 | 71.2 | 52.0 | 56.1 | 71.2 |
| Dep-Guided LSTM (Jie & Lu, 2019) | 42.8 | 41.7 | 49.8 | 56.0 | 71.1 | 71.6 | 52.4 | 55.8 | 70.7 |
| Focal Loss (Lin et al., 2017) | 38.6 | 42.9 | 50.7 | 58.8 | 74.6 | **76.0** | 52.6 | 58.5 | **72.7** |
| cRT (Kang et al., 2019) | 44.8 | 47.4 | 58.8 | 60.1 | 68.8 | 68.5 | 57.6 | 58.9 | 71.4 |
| $\tau$ - Normalization (Kang et al., 2019) | 34.3 | 35.6 | 50.9 | 53.8 | **82.7** | 68.3 | 53.3 | 52.5 | 67.6 |
| LWS (Kang et al., 2019) | 34.3 | 35.6 | 61.2 | **60.2** | 76.8 | 71.7 | 58.2 | 56.9 | 67.5 |
| TDE (Tang et al., 2020b) | 34.3 | 33.9 | 61.5 | 59.7 | 77.4 | 73.3 | 58.5 | 56.5 | 72.0 |
| Our (Glove) | **47.1** | **49.7** | 64.3 | 59.9 | 80.5 | 73.3 | **63.5** | 60.2 | 71.5 |
| BERT (Devlin et al., 2019) | 47.6 | 48.9 | 67.8 | 67.5 | 84.5 | 76.8 | 66.5 | 65.1 | 77.6 |
| Roberta (Liu et al., 2019a) | 47.6 | 47.7 | 73.7 | 72.6 | 86.0 | **80.9** | 70.2 | 68.6 | **80.4** |
| BERT+GCN (Wadden et al., 2019) | 45.2 | 47.5 | **78.9** | **77.5** | **88.2** | 80.3 | 73.1 | 71.3 | 80.3 |
| Ours (BERT) | **61.9** | **63.2** | 76.5 | 76.6 | 85.3 | 80.7 | **74.9** | **74.4** | 80.1 |

Table 13: Evaluation results on the MAVEN dataset for the event detection.

| Model | Few | | Medium | | Many | | Overall | | |
|---|---|---|---|---|---|---|---|---|---|
| | MR | MF1 | MR | MF1 | MR | MF1 | MR | MF1 | Micro F1 |
| BiLSTM | 36.5 | 40.7 | 78.3 | 79.9 | 80.4 | 82.3 | 67.1 | 69.5 | 83.0 |
| BiLSTM+CRF | 43.4 | 46.8 | 79.0 | 79.8 | 82.3 | 83.0 | 69.6 | 71.1 | 83.3 |
| CGGCN (Zhang et al., 2017) | 49.7 | 51.7 | 81.8 | 80.8 | 82.6 | 82.1 | 73.1 | 73.0 | 83.6 |
| Dep-Guided LSTM (Jie & Lu, 2019) | 44.7 | 45.4 | 76.5 | 78.2 | 75.9 | 78.9 | 67.8 | 69.3 | 83.0 |
| Focal Loss | 45.4 | 51.5 | 78.6 | 81.3 | 85.4 | 87.2 | 70.3 | 73.8 | **84.9** |
| cRT | 49.7 | 55.4 | 78.4 | 81.3 | 82.1 | 85.0 | 71.0 | 74.6 | **84.9** |
| $\tau$ - Normalization | 21.1 | 26.7 | 60.0 | 68.5 | 74.4 | 80.0 | 51.0 | 58.4 | 76.0 |
| LWS | 33.3 | 38.7 | 77.6 | 79.7 | 81.6 | 81.7 | 65.9 | 68.7 | 83.0 |
| TDE (Tang et al., 2020b) | 39.8 | 36.2 | 83.3 | 78.0 | 87.8 | 85.2 | 71.9 | 67.4 | 79.1 |
| Our (Glove) | **60.4** | 57.4 | 86.8 | 82.2 | 89.1 | 86.6 | **79.8** | 76.0 | 84.6 |
| BERT (Devlin et al., 2019) | 38.1 | 39.3 | **86.3** | 84.0 | **90.6** | 88.3 | 73.5 | 72.3 | 86.2 |
| Roberta (Liu et al., 2019a) | **43.1** | 43.8 | **86.3** | **84.1** | 90.2 | 88.4 | **75.0** | **73.6** | **86.2** |
| BERT+GCN (Wadden et al., 2019) | 40.9 | 42.0 | 86.0 | 83.6 | 90.0 | **88.6** | 74.1 | 72.8 | 86.1 |
| Ours (BERT) | **43.1** | **45.0** | 80.7 | 80.5 | 88.3 | 87.6 | 71.3 | 71.7 | 85.0 |

Table 14: Evaluation results on the NYT24 dataset for the relation extraction task.

| Model | Few | | Medium | | Many | | Overall | | |
|---|---|---|---|---|---|---|---|---|---|
| | MR | MF1 | MR | MF1 | MR | MF1 | MR | MF1 | Micro F1 |
| C-GCN (Zhang et al., 2017) | 24.0 | 26.7 | 60.6 | 65.5 | 58.8 | 59.1 | 51.2 | 52.6 | 74.4 |
| Focal Loss (Lin et al., 2017) | 52.0 | 48.3 | 78.0 | 74.8 | 63.5 | 64.0 | 62.9 | 61.9 | 74.5 |
| cRT (Kang et al., 2019) | 66.0 | 24.2 | 75.2 | 58.5 | 64.6 | 61.9 | 65.6 | 50.5 | 68.0 |
| $\tau$ - Normalization (Kang et al., 2019) | 40.0 | 40.0 | 59.0 | 64.7 | 57.1 | 57.7 | 53.5 | 54.6 | 74.3 |
| LWS (Kang et al., 2019) | 40.0 | 40.0 | 59.0 | 64.7 | 57.1 | 57.6 | 53.5 | 54.6 | 74.3 |
| TDE (Tang et al., 2020b) | 60.0 | 57.1 | 66.6 | 65.2 | 60.1 | 60.2 | 61.0 | 60.2 | 70.0 |
| Ours (Glove) | **68.0** | **68.6** | 67.0 | 70.5 | 59.3 | 60.3 | 65.3 | 63.6 | 72.3 |

