# OpenReview forum: "Counterfactual Thinking for Long-tailed Information Extraction"
_ICLR.cc/2021/Conference — Reject_

### Official Review · AnonReviewer3 · 2020-10-25
**Official Blind Review #3**

**Rating:** 3
**Confidence:** 4

**Review:**

This paper propose a counterfactual approach to improve the performance in information extraction tasks and in particular on the rarer classes. While this is an important research problem, I think the paper has the following issues:

- The overall  approach is best described as a form of a data augmentation based on the model predictions. Steps 1 to 3 is essentially finding out which part of the input had the most impact on the model prediction by comparing the impact of a token being masked at random, and then steps 4 and 5 use this info to improve the prediction. This is quite commonly done in the context of model interpretation, e.g. finding out which words matter the most in a prediction was done in the work on LIME: http://sameersingh.org/files/papers/lime-kdd16.pdf , and by many others that followed this up, e.g. finding out which words are the most important via removal: https://arxiv.org/abs/1804.07781. The latter is actually doing what the first three steps do of the proposed approach. Steps 4 and 5 essentially use this output which presumably captures the contextual information and combine it with the contextual word representation, which is a pretty standard thing to do. However this existing work in NLP has been ignored.

- The causal framing in Figure 1 appears flawed. There is no good reason to assume that the NER tag "causes" the trigger representation and not the other way around. However this is a fundamental assumption here. Also, why is not the contextual word representation "causing" the NER tag? Furthermore, there is no causality inferred here; the model output is post-processed, and the causality is not assessed in any way.

- Few-shot learning has been investigated in the context of information extraction. Here are some papers: https://www.aclweb.org/anthology/D19-1649/ https://www.aclweb.org/anthology/P19-1589/ .

- The paper is hard to follow. What is v_r in equation 3? In section 2.2, it should be explicit that the tokens are not replaced, but removed. Note that replacement with semantically equivalent words has been explored: http://sameersingh.org/files/papers/sears-acl18.pdf

- There are some vague statements about the novelty vs previous work by Tang et al, but no explicit statement made in the model description. Eventually it seems to be about considering the syntax (page 7), but this is only used to just select which words to remove as far as I can tell and the GNN over dependency trees which is based on previous work.

- While it is stated that the results of previous work are reproduced, then it is stated that they were not really. While this is not necessarily the authors' fault (reproducibility can be difficult sometimes), combined with the uncommon choice of metrics and the lack of any comparison to previously reported results in the literature, means that it is impossible to understand where the presented method stands.

---

> ### Author Response · Authors · 2020-11-25
> **We would like to thank the reviewer’s thoughtful comments and the recommendation of some interesting papers in the field of model interpretation. Also, we are sorry if our original version has led to some misunderstandings and we would like to give some clarifications. We have improved the presentations in our revised paper following the reviewer's helpful comments and also give clarifications for these unclear points as follows.**
>
> **Point 1:**
> Referee: Steps 1 to 3 is essentially finding out which part of the input had the most impact on the model prediction by comparing the impact of a token being masked at random, and then steps 4 and 5 use this info to improve the prediction. This is quite commonly done in the context of model interpretation.
>
> **Reply**: We agree that the proposed CFIE masks the tokens that have the most impact on the prediction Steps 1 to 3. However, our masking operations are based on the dependency structure or its shortest path for different IE tasks. The new sequence will be used to generate counterfactuals from the test data during the inference stage. To the best of our knowledge, such a way has not been explored in literature so far.
>
> We use the generated counterfactuals to eliminate the spurious correlations in step 4 and 5 in the perspective of causal inference. To the best of our knowledge, the method we proposed in Step 5 has not been proposed in literature so far, i.e., a more robust inference approach for the long-tailed IE tasks.
>
> **Point 2:**
> Referee:This is quite commonly done in the context of model interpretation, e.g. finding out which words matter the most in a prediction was done in the work on LIME: http://sameersingh.org/files/papers/lime-kdd16.pdf , and by many others that followed this up, e.g. finding out which words are the most important via removal: https://arxiv.org/abs/1804.07781. The latter is actually doing what the first three steps do of the proposed approach.
>
> **Reply**: We have carefully studied the papers and we agree that our work is related to model interpretation to some extent as generally causal models are interpretable. However, we believe that our work is different from the two works both in the motivation and the methodologies.
>
> The two interesting papers referred by the reviewer are LIME http://sameersingh.org/files/papers/lime-kdd16.pdf  and  “Pathologies of Neural Models Make Interpretations Difficult:” (https://arxiv.org/abs/1804.07781). LIME aims to identify an interpretable model over interpretable representation that is locally faithful to the classifier. The focus of LIME is to find out the clues that can interpret a prediction to decide if one should trust the prediction. On the contrary, we don’t need to find out these clues to explain a model. We rely on the dependency structure to mask the context/entities to generate counterfactuals. In other words, we have the prior that these clues play an important role for the classification. The intuition behind is that the dependency tree has been proven effective for the IE tasks. Further, compared with LIME, our aim of generating these counterfactuals is to mitigate the spurious correlations for the long-tailed IE rather than explain the classifier to help the user in an informed position.
>
> The second paper: https://arxiv.org/abs/1804.07781 (named input reduction) aims to find out the most important input features by iteratively removing the least important words. The authors conduct experiments on question answering, Stanford Natural Language Inference (SNLI), and VQA tasks. It is very interesting that the model is still able to make the same prediction with high confidence based on the remaining words. However, we argue that our method in steps 1 to 3 is very different from the input reduction used in https://arxiv.org/abs/1804.07781. The key differences are listed as follows.
> 1) The goals of these two methods are different. Our aim is to generate the counterfactuals based on a dependency tree by masking tokens or entities, and we are not trying to find the most important words by beam search as the input reduction did. We highlight this in Section 3.2.
> 2) The methodologies are also different. We have the prior knowledge that some context/entities selected by the dependency structure may be important to the prediction. As described in Section 3.2 of our paper, we mask the relevant words to generate counterfactuals in the perspective of causal inference. However, the input reduction method referred by the reviewer relies on some predefined rules such as the beam search to find these less important words and then remove them step by step.
>
> 3) The tasks that the two methods applied to are totally different. The input reduction referred by the reviewer is applied in QA and SNLI tasks, which require semantic alignment between two inputs such as context and query. We believe the reason why the reduced nonsensical words still obtain high confidence is that the model is able to make accurate alignment based on the given unreduced context. Such unchanged contexts are the support document in SQUAD, images for VQA,  and premise for SNLI.  However, in the case of IE tasks, it is infeasible to apply this method here as we count on only one given sentence to make predictions. We also show that using reduced nonsensical words may not yield accurate predictions in Section 4.5 and Appendix A4.1, A4.2.

---

> > ### Author Response · Authors · 2020-11-25
> > **Continue**
> >
> > **Point 3: **
> >
> > **Referee**: steps 4 and 5 essentially use this output which presumably captures the contextual information and combine it with the contextual word representation, which is a pretty standard thing to do
> >
> > **Reply**: To the best of our knowledge, our classifier proposed in Step 5, which is inspired by total direct effect in causal inference, is the first attempt that is able to make robust predictions for the IE tasks.  Such a way can be better described as combining the main effect logits (Y_{x_i) - Y_{x^*_i}) as well as the counterfactual logits (W_{XY} x^*_i) of an entity to yield a more robust prediction. Details are given in the Eq. (8). We also give the sensitivity analyses for the parameter \alpha in Figure 9 of Section 4.5 and Figure 12 of Appendix A.4.3. The parameter indicates the importance of the W_{XY} x^*_i.
> >
> > **Point 4:**
> >
> > Referee:The causal framing in Figure 1 appears flawed. There is no good reason to assume that the NER tag "causes" the trigger representation and not the other way around. However this is a fundamental assumption here. Also, why is not the contextual word representation "causing" the NER tag? Furthermore, there is no causality inferred here; the model output is post-processed, and the causality is not assessed in any way.
> >
> > **Reply**: Previous studies have shown that NER tags are effective to the trigger classification for the ED task(Liu et al, 2019) and it supports our design in SCM that NER tag can be a cause for the prediction.  The experiments in Section 4.5 also prove that NER tags help the trigger prediction. For the ED task in our setting, NER tags are given and so there is no need to infer the causality between the contextual word representations and the NER tag. Jointly training NER and ED is beyond the scope of this study and we will leave it as our future study.
> >
> > We have added the above explanations in our revised paper.
> >
> > Liu, Shulin, et al. "Event Detection without Triggers." Proceedings of the 2019 Conference of the North American Chapter of the Association for Computational Linguistics: Human Language Technologies, Volume 1 (Long and Short Papers). 2019
> >
> > **Point 5:**
> >
> > **Referee**: Few-shot learning has been investigated in the context of information extraction. Here are some papers: https://www.aclweb.org/anthology/D19-1649/ https://www.aclweb.org/anthology/P19-1589/ . Only token-level tasks are explored, but often in IE we want to infer relations between tokens, which this work doesn't apply to.
> >
> > **Reply**: We have studied the reviewer’s recommended papers about few shot learning. We agree that the few shot learning has been applied to IE tasks. However, our CFIE aims to improve the tail class (few training instances)  classifications by mitigating the spurious correlations caused by the head class in an imbalance dataset. Although the long-tailed IE shares some similarities with the few shot learning with a very limited number of training instances, the objectives of the two tasks are quite different. Few shot learning aims to handle new classification tasks with only a handful of training instances.
> >
> > We are confused on the point “Only token-level tasks are explored, but often in IE we want to infer relations between tokens, which this work doesn't apply to”. Besides NER and ED, our proposed CFIE is also applied to the relation extraction task, which aims to predict the relations between entities that may consist of multiple tokens.
> >
> > We have added the two references mentioned in the review comments, as well as the above explanations in our revised version.

---

> > > ### Author Response · Authors · 2020-11-25
> > > **Continue2**
> > >
> > > **Point 6: **
> > >
> > > **Referee**:
> > > The paper is hard to follow. What is v_r in equation 3? In section 2.2, it should be explicit that the tokens are not replaced, but removed. Note that replacement with semantically equivalent words has been explored: http://sameersingh.org/files/papers/sears-acl18.pdf
> > >
> > > **Reply**:  v_r is the representation of the dependency relation between two tokens in the sentence. We add the explanations in our revised version.
> > >
> > > For the comments “In section 2.2, it should be explicit that the tokens are not replaced, but removed”: In this paper, we only mask the tokens with an embedding vector configured as zeros based on the dependency structure  to generate counterfactuals rather than removing them. We have clarified it in our revised version.
> > >
> > > For reference paper SEARs
> > > “http://sameersingh.org/files/papers/sears-acl18.pdf”:
> > > We have carefully studied the SEARs that is recommended by the reviewer.  We argue that SEARs is very different from ours both in motivation and methodology: 1) SEARs can be considered as a data augmentation technique to enhance the model’s robustness  by replacing the words with semantically equivalent words during the training stage. However, we only generate the counterfactuals from the test data during the inference stage and don’t use these counterfactuals to train the model parameters. Our method is orthogonal to data augmentation techniques used in SEARs.  2) we aim to generate counterfactuals to mitigate the side effect of IE tasks by masking the word based on the dependency structure, while the SEARs referred by the reviewer aims to detect bugs in black-box state-of-the-art models by inducing adversaries on many instances.
> > >
> > > **Point 7:**
> > >
> > > **Referee**:There are some vague statements about the novelty vs previous work by Tang et al, but no explicit statement made in the model description. Eventually it seems to be about considering the syntax (page 7), but this is only used to just select which words to remove as far as I can tell and the GNN over dependency trees which is based on previous work.
> > >
> > > **Reply**: We agree that both our approach and the model proposed by Tang et al., 2020 attempt to address the long-tailed bias issues from NLP and CV perspective, respectively. However, we differ from Tang et al., 2020 in three aspects : 1) the counterfactuals of our CFIE are generated based on the dependency structures of a sentence, while the ones in Tang et al., 2020 are generated by simply setting the representations of the objects as NULL.  2) We propose a more flexible inference approach for the three IE tasks compared with Tang et al., 2020. We show in experiments that directly applying the Eq. (7), which is similar to the TDE approach proposed by Tang et al., 2020, is unable to yield good performance on NER and ED.  We therefore introduce the W_{XY} x^* as shown in the Eq. (8) for the two tasks, which can be considered as another key contribution of our work. 3) the proposed SCMs are very different from Tang et al., 2020, which involves a mediator Z that is caused by X, while our SCMs simply consider the Z variables as given linguistic features.
> > >
> > > We use GNN here as a graph encoder to better capture the dependency structure. We agree that such an application is not our contribution. Our CFIE is a general framework for the IE tasks and other graph encoders can also be integrated in it to benefit the overall performance.
> > >
> > >
> > > **Point 8:**
> > >
> > > **Referee**: While it is stated that the results of previous work are reproduced, then it is stated that they were not really. While this is not necessarily the authors' fault (reproducibility can be difficult sometimes), combined with the uncommon choice of metrics and the lack of any comparison to previously reported results in the literature, means that it is impossible to understand where the presented method stands.
> > >
> > > **Reply**: We thank the reviewer’s understanding of the difficulty in re-implementation. In our revised version, we have clarified this point to avoid any confusion.  We have also released our code, detailed instructions, and pre-trained model to let the interested readers reproduce the results.

---

### Official Review · AnonReviewer1 · 2020-10-26
**Interesting idea with comprehensive experiments**

**Rating:** 6
**Confidence:** 4

**Review:**

This paper proposes a novel model integrating both causal inference and structure-aware counterfactual training to enhance the long-tail performances of information extraction. The causal mechanism considers a structured causal model that takes into account all possible cause-effect relations for the final predictions, including contexts, target representations, POS tags, NERs, etc. They also implement counterfactual training strategy by selecting the most important factors and wipe off the side effects to enhance the long-tail situations.

The strengths of the paper includes:
1. In general, this paper is well-written and easy to follow. The motivation and the structure are clear.
2. The ideas of both structured causal model and structure-aware counterfactual training are interesting.
3. Extensive experiments are conducted to demonstrate the effect of the whole model and each component. It is interesting to see how different generation of counterfactual examples using dependency structure affect the final performance.

Some improvements could be made:
1. If structure is considered, why not try to mask on some dependency relations? It will be interesting to see the difference between masking words and relations.
2. What is the effect of using (5) instead of (4) in terms of the experimental result? Have you conduct such comparison experiment? And how sensitive is $\alpha$ to the final performance?
3. It is also better to demonstrate some qualitative examples on which factors are most important for NER, RE and ED.

---

> ### Author Response · Authors · 2020-11-25
> **Thanks for your thoughtful comments.**
>
> **Q1**: For masking the dependency relations.
>
> **A1**: We agree that it will be more interesting to see the difference between masking words and relations. In our revised version, we have conducted the experiments for the RE task and have added the analyses in Table 9 Appendix A.5.
>
> **Q2**:  For the effect of using (6) instead of (5) (The numbers are (8) and (7) in  the original submission).
>
> **A2**: Eq (8) is designed for the NER and ED tasks to strengthen the entity itself representations. We show the results in the Table 9 and Table 12. When \alpha is set to 0, it stands for Eq (7). We observe that using (8) instead of (7) for the RE task can improve the NER and ED on four datasets.
>
> **Q3**: The sensitivity of α to the final performance.
>
> **A3**: In our revised version, we have added the experiments to analyze the sensitivity of the parameter α in Figure 9 of Section 4.5 and Appendix A.4.3.
>
> **Q4**: Qualitative examples to demonstrate which factors are most important for NER, RE and ED tasks.
>
> **A4**: Using qualitative examples to demonstrate the most important factors for NER, RE and ED tasks is an interesting idea. We have added such a visualized demonstration and analyses for each task in Section 4.5 and Appendix A.4.1 and A.4.2.

---

### Official Review · AnonReviewer2 · 2020-10-28
**Counterfactuals for Long-Tail IE with help of GNN and dependency tree**

**Rating:** 7
**Confidence:** 3

**Review:**

#### Update after author response and other reviewers comments:
I think after the addition of new evaluation (Appendix A.5) on the aspect of counter-factual and other comments made by reviewers I'll stick to my score. Also, I liked the idea of applying counter-factual to long-tailed distribution IE problem.

### Summary
Model performance suffers because of spurious relations present in the data, this is particularly true for long-tailed scenarios. To overcome this challenge authors introduces counterfactual thinking to IE. To learn the main effect and ignore the spurious relations (side effects) the paper proposes structured causal model (SCM) with syntax structures using GNN on sentence dependency tree. Extensive experiments on multiple tasks with different datasets shows significant performance improvement.

### Strong Points
 1. Application of counterfactual along with utilization of syntax structures for long-tailed IE is novel and aptly motivating.
 2. I found the idea of measuring the main effect and a way to ignore the "side effects" very interesting.
 3. Extensive experiments provides ample evidence that reducing spurious relations leads to improved performance.

### Weak Points
 1. How incorporation of the 3rd component (W_{x}X^∗) in Eqn. 5 measures the effects of the entity is not clear to me. X^* is already taken into account when calculating Y_{x^∗}(S), is it not?

### Other comments
 1. The phrase "**replace** the tokens along the shortest path of the two entities of the relations and generate a new sequence S^∗" was not really clear to me from the paper. Could you please explain this part such as replaced with what?
 2. What prompted the authors to consider ≤ 4000 as few scenario for "OntoNotes5.0" dataset whereas for all other scenarios it's either 30 or 100? I feel **4000** is too large for it to be considered "Few" for a class.
 3. It's curious to find that GloVe based approaches outperformed BERT based approaches by a large margin in ATIS dataset in NER task (Table 1) and MAVEN dataset for ED task.

---

> ### Author Response · Authors · 2020-11-25
> **Thanks for your thoughtful comments.**
>
> **Q1**: For the  3rd component (W_{x}X^∗) in Eq. 8 (The number in the original version is 5)
>
> **A1**: In the revised version, we use W_{XY}  x^*_i  to replace the 3rd component in the Eq.8. The Eq.8 is used to calculate the direct causal effect of X ->Y for the NER and ED tasks.  Here we rely on two examples given in Figure 1 to describe how we use the Eq.8 and W_{XY}  x^*_i.
>
> In the ACE2005 dataset,  the token “killed” is involved in 511 training instances of the type “Life:Die” and only 10 instances of the event “SW:Quit”. A traditional model based on correlations tends to predict a trigger “killed” as the event “Life:Die”, no matter what context is around the trigger. For example,  the trigger word “killed” in the sentence “the program was killed” may be incorrectly predicted as “Life:Die”  type. Our aim is to predict the correct event type of the token “killed” in “the program was killed”, i.e., the event type  “SW:Quit”.
>
> We train the model in Step 1 and obtain the prediction logits Y_x in Step 2. Then we generate the counterfactuals “the __ was killed” by masking the token “program” based on the dependency tree, and obtain the prediction logits Y_{x^*} with the same model. Then we can compute the Y_x - Y_{x^*} in Eq 8, which mainly refers to the interventional causal effect of the masked token  “program” on the prediction results. The intuition behind is that the token “program”, which resides in the 1-hop distance to the target token “killed” on the dependency tree, may be the important clue for the event classification. Detailed discussions are given in Appendix A 4.1.
>
> However, such a computation ( Y_x - Y_{x^*})  may weaken expression of the trigger word itself “killed”, as the both Y_x  and Y_{x^*} contains its information. We therefore need to strengthen the trigger’s impact by introducing W_{XY}  x^*_i in Eq 8, as a trigger also plays an important role for the classification for the ED task. We use the parameter \alpha to control the contribution of this component to the overall performance. We discuss the importance of trigger in Appendix A 4.1, as well as the sensitivity of the parameter \alpha in Section 4.5.
>
>
> **Q2**: For the "replace the tokens along the shortest path of the two entities of the relations and generate a new sequence S^∗"
>
> **A2**: For the RE task, we first extract the shortest path between two entities in a dependency tree. Then we mask the tokens involved in the shortest path to obtain a new sequence, which will be fed to our model to generate the representations S^*.   We simply set the embedding of the token to be masked as the zero tensor.
>
>
> **Q3**:  For the setting of few-shot setting on OntoNotes5.0 dataset.
>
> **A3**: The scale of the OntoNotes5.0 dataset  is much larger than other datasets. It is very different from the ACE2005 dataset.  The number of training samples for each class in ACE2005 dataset ranges from 6 to 1343. However, the number of instances for each class in OntoNotes5.0 are 374, 1671, 1815, 2216, 3102, 3129, 3180, 4143, 4154, 4521, 9227, 9983, 14553, 15207, 27374, 37392, 41359, 57156. So we categorized the first 7 classes as the Few type as the number of the largest Few class (3180) is about 8% of the head class (57156).  We will try to obtain a more reasonable split.
>
> **Q4**: For the BERT results:
>
> **A4**: We are also surprised that the Glove-based approaches perform better than BERT-based models on ATIS and MAVEN datasets in terms of long-tailed metrics MR and MF1. However, the results in terms of Micro F1 are much more intuitive. As shown in Table 10 in the Appendix, the former performs worse than the latter in terms of Micro F1. One possible reason is that the pre-trained BERT/Roberta encoders may still introduce spurious correlations for the entity classification, resulting in lower MR and MF1.  A similar issue is also discussed in a very recent paper (Yue et al., 2020), where pre-trained models are a confounder that may cause unexpected high correlations.  We will add the above explanations in the paper.
>
> Yue, Zhongqi, et al. "Interventional few-shot learning." Advances in Neural Information Processing Systems 33 (2020).

---

### Official Review · AnonReviewer4 · 2020-10-29
**This paper first proposes an SCM-model based on masked contextualized word representation learning to capture the direct slide effect; Second, counterfactuals based on syntax structure is computed to invervene on the model and an IE classification model are proposed to solve the slide effect and counterfactual rebalancing problem for unbiased IE prediction.**

**Rating:** 5
**Confidence:** 5

**Review:**

The novelty of the paper seems to be in application of the counterfactual analysis to address the long-tailed IE issues, which might be interesting to the IE researchers. Overall, more theory about the counterfactual generation for IE task should be added, for this is what the novelty of the paper; also, for the rebalancing learning for slide effect and counterfactual, the theory appears to be not enough. The weak of this work is the theoretical and conceptual underpinnings of the proposed methodology.

Here are my major concerns for  the paper:
Q1. The main investigation of the paper is the unstructured text; however, I can see no discussion in the methodology part about how author represent unstructured text in SCM model (representation learning). Is it unimportant to mention or the SCM model definition is always the same no matter what sources of the data?
Q2. Section2.2. Counterfactual Generation, as this forms the main contribution of this work, still no useful information is given in this section, but only a do-calculus is given. Since this remains at the basic concept of causal inference, how can the paper combine this with IE task?
Q3. Equation (3) u_i appears to be the representation of the i_th position of a sentence, which I think is important notion across the whole paper, however, in the later illustration after equation(3), u_i didn’t appears again even once. Why? I am doubtful about the reproducibility of the paper.
Q5. How author design the optimization function? How equation (5) (3) been used?
Q4. As for the experiment, author use MR and MF1 as the evaluation metrics, since this work can measure causal effect. More evaluation regarding the causal effect should be added.

---

> ### Author Response · Authors · 2020-11-25
> **Thanks for your thoughtful comments. We have added more details about the mathematical formulations of the counterfactual generation as well as the proposed causal effect estimation method in Section 3.2 and 3.3. Due to space limitation, more details about the derivations are given in Appendix A.1.**
>
> Q1: For representing unstructured text in SCM model:
>
> **Reply: ** We use BiLSTM or pre-trained BERT and RoBerta to output the contextualized representation of the unstructured text. In our case, we use the same SCM for these encoders. We add the above explanations in the paper to make it more clearly.
>
> Q2: For combining counterfactual generation with IE task:
>
> **Reply: ** We have detailed the counterfactual generation in Section 3.2 in our revised version, including the definition of  interventions, the formulation of counterfactuals, as well as the dependency-guided counterfactuals generation.
>
> In our revised version, we have also added the detailed description about how our model combines the IE tasks in the last paragraph of Section 3.2 and in Section 3.3.
>
>
> Q3: For the u_i and the reproducibility of the paper
>
> **Reply: ** In our original version, u_i refers to the token-level contextualized representations, which will be used to obtain the prediction logits expressed by Y.  In our revised version, we replace u_i to x_i to make it more consistently.
>
> For the reproducibility,  we have uploaded our code, preprocessed data, as well as the pre-trained model for the NER task. We also write a step-by-step instruction to facilitate the interested readers reproduce the results. We believe anyone can reproduce similar results on the ATIS dataset for the NER tasks as shown in Table 5.
>
> Q4: For the optimization function and usage of equations.
>
> **Reply: ** In the training stage, we have four losses, including the loss of main logits, as well as three auxiliary losses that are computed based on S->Y, Z->Y, and X->Y, respectively.  We add the mathematical formulation in the Equation (2).
>
> For the usage of Equation (4) and (8) (The numbers are 3 and 5 in our original version): Both equations are used for the NER and ED tasks. Equation (4) is used for information aggregation of  the i-th token based on a dependency tree. We use Equation (8) to compute the direct causal effect between X->Y to make robust predictions for NER and ED tasks.
>
> Q5: For the more evaluations on the causal effect.
>
> **Reply: ** We have added more experiments to measure direct causal effect for the RE task. We list results in the Appendix A.5. We measure the average causal effect from different factors to the ground truth, including S, X, NER, POS, tokens along the shortest path between the subject and object, and dependency relations.

---

### Decision · Program_Chairs · 2021-01-07
**Final Decision**

**Decision:**

Reject

**Comment:**

This paper introduces an architecture based on structured causal model for long-tailed IE tasks. It incorporates the dependency tree structure of the sentence using a GCN for learning the representations. The key idea is to use counterfactual reasoning to help with the inference in attempt to reduce the impact of spurious relations. There are some concerns about the presentation of this paper. While the high level idea is reasonably clear and well motivated, the paper is quite messy with the notations and technical details.
How to use the causal effect estimation for the final prediction is not explicitly explained except for in Figure 1.
For the experiments on ED and NER, it is unclear if they assume the trigger or span is given. The method seems to need the span information to make the prediction. If span is given, this is a different set up that is much simpler compared to traditional ED and NER where the span or trigger needs to be detected as well.
There are also some question regarding the difference between this work and the prior work on using causal reasoning for improving prediction (the TDE work). One difference is the additional term in equation 8(updated version), which appears to be useful empirically, but the motivation is rather hand wavy and needs more clarification.
Overall, there are some useful ideas, but the overall novelty does not particularly stands out, and the presentation of the paper made somewhat straight forward ideas more convoluted than necessary.